

# Ensemble learning approach for distinguishing human and computer-generated Arabic reviews

Fatimah Alhayan[1] and Hanen Himdi[2]

[1] Department of Information Systems, College of Computer and Information Sciences, Princess Nourah Bint Abdulrahman University, Riyadh, Saudi Arabia
[2] Computer Science and Artificial Intelligence Department, College of Computer Science and Engineering, University of Jeddah, Jeddah, Saudi Arabia

## ABSTRACT

While customer reviews are crucial for businesses to maintain their standing in the marketplace, some may employ humans to create favorable reviews for their benefit. However, advances in artificial intelligence have made it less complex to create these reviews, which now rival real ones written by humans. This poses a significant challenge in distinguishing between genuine and artificially generated reviews, thereby impacting consumer trust and decision-making processes. Research has been conducted to classify whether English reviews were authored by humans or computers. However, there is a notable scarcity of similar studies conducted in Arabic. Moreover, the potential of ensemble learning (EL) techniques, such as soft voting, to enhance model performance remains underexplored. This study conducts a comprehensive empirical analysis using various models, including traditional machine learning, deep learning, and transformers, with an investigation into ensemble techniques, like soft voting, to classify human and computer-generated Arabic reviews. Integrating top logistic regression (LR) and convolutional neural network (CNN) models, it achieves an accuracy of 89.70%, akin to AraBERT's 90.0%. Additionally, a thorough textual analysis, covering parts of speech (POS), emotions, and linguistics reveals significant linguistic disparities between human and computer-generated reviews. Notably, computer-generated reviews exhibit a substantially higher proportion of adjectives (6.3%) compared to human reviews (0.46%), providing crucial insights for discerning between the two review types. The results not only advance natural language processing (NLP) in Arabic but also have significant implications for businesses combating the influence of fake reviews on consumer trust and decision-making.

# INTRODUCTION

In today's digital age, e-commerce platforms play a crucial role in consumer decision-making. These platforms are heavily influenced by online reviews, which wield substantial power in shaping purchasing decisions and building consumer trust (*Fernandes et al., 2022*; *Fahrozi et al., 2022*). However, the integrity of this feedback system is increasingly

Corresponding author
Fatimah Alhayan,
fnalhayan@pnu.edu.sa

threatened by manipulation tactics. The proliferation of fake reviews, designed to artificially boost product ratings or undermine competitors (*Choi et al., 2017*; *Zhu et al., 2023*), significantly compromises the reliability of online feedback mechanisms.

The emergence of large language models (LLMs), such as OpenAI's GPT (Generative Pre-trained Transformer) (https://openai.com/about), which have gained prominence in recent years, exacerbates this issue by facilitating the creation of customer reviews at a reduced cost (*Gambetti & Han, 2023*). These models possess sophisticated text generation capabilities capable of producing human-like reviews (*Li et al., 2024*), blurring the distinction between computer-generated and human-written content and enabling deception and misinformation (*Barman, Guo & Conlan, 2024*). Misinformation from LLMs is more challenging for both humans and detection systems to identify than similar human-written content, making it more deceptively styled and potentially more harmful (*Chen & Shu, 2023*).

These advancements pose significant challenges for platforms like TripAdvisor, Yelp, and Google, which frequently contend with an influx of fake reviews aimed at manipulating public perception (*Adelani et al., 2020*). Such reviews can be positive, designed to boost business profits, or negative, intended to undermine competitors or achieve political objectives. Consequently, these deceptive practices severely undermine the reliability of review systems, fostering doubt and uncertainty among consumers in the marketplace.

Additionally, LLMs generate text based on learned patterns from diverse sources, causing confusion among consumers. For example, a phrase like "compact design" might seem positive but could actually mean different things. It might be used to praise a product's small size or to criticize its lack of features. This ambiguity makes it difficult for shoppers to grasp the genuine qualities of a product, potentially leading to erroneous purchasing decisions. Therefore, online reviews on digital platforms significantly influence consumer decisions and trust in products and services (*Gambetti & Han, 2024*).

Addressing vulnerabilities in LLMs, especially their ability to generate deceptive customer reviews that appear authentic, is crucial for maintaining trust in digital platforms. Current research primarily focuses on distinguishing between human and computer-generated text using machine learning (ML), deep learning (DL), and/or Transformers (TF) models (*Bader et al., 2023*; *Mitrović, Andreoletti & Ayoub, 2023*; *Gambetti & Han, 2023*; *Buscaldi & Liyanage, 2024*; *Chen & Shu, 2023*). However, there remains a significant gap in exploring the potential of alternative techniques such as ensemble learning (EL) methods like soft voting to enhance the accuracy of these detection models. Additionally, most studies have concentrated on languages other than Arabic, limiting our understanding and effectiveness in detecting fake reviews within this specific linguistic context. The novelty of this study lies in addressing these gaps by focusing on Arabic, thereby advancing the field and improving the reliability of identifying fake reviews across diverse languages and platforms.

Therefore, this article proposes an enhanced model for detecting computer-generated reviews by evaluating the performance of various artificial intelligence (AI) models, including ML, DL, and TF, and comparing them with EL techniques on Arabic reviews.

These reviews were generated using two LLMs: GPT, a specific type of large language model developed by OpenAI (https://openai.com/about), and ULMFit (Universal Language Model Fine-Tuning), a technique for refining a pre-trained language model to perform specific tasks (*Howard & Ruder, 2018*). Both models can generate text that closely resembles human writing based on provided prompts.

The key contributions of the research are as follows:

- The research represents the first work to classify human-generated and computer-generated Arabic reviews by compiling a diverse range of AI models, including ML, DL and TF models.
- The research proposes ensemble learning and hybrid techniques aimed at boosting the performance of the models.
- The research conducts a thorough Arabic textual analysis, comparing the reviews generated by humans to those generated by computers, aiming to identify unique linguistic patterns.

The rest of the article is organized as follows: "Related Work" discusses the related work. "Methodology" provides a mathematical formulation for the proposed algorithm, including a detailed methodology used in this study, consisting of data collection, preprocessing, feature engineering, and model compilation. Textual analysis was performed in "Textual Analysis". "Results" and "Discussion" provide results and discussion on results, respectively. Finally, "Limitations and Future Works" and "Conclusions" conclude the work.

## RELATED WORK

Detecting fake reviews has gained significant importance across various domains. This is particularly crucial not only for promoting products or services in e-commerce but also for e-government and health sectors. In these contexts, the reliability of reviews plays a vital role in promoting or evaluating products or services, building consumer trust, and influencing consumer decision-making. A growing issue revolves around the utilization of LLMs to generate deceptive reviews. The capacity of these models to generate authentic-looking content on a large scale introduces new challenges for online platforms in recognizing and addressing such deceptive reviews (*Gambetti & Han, 2024*). Nevertheless, the majority of existing studies primarily concentrate on detecting fake English reviews.

*Bader et al. (2023)* is a pioneering effort in detecting fake reviews generated with ChatGPT. The study analyzed 6,013 English reviews of the Facebook mobile app, consisting of 4,000 real reviews from Google Play and 2,013 fake reviews generated by ChatGPT. Researchers employed lexical features like text length, stopword percentage, and sentiment polarity to train decision tree (DT), support vector machine (SVM), and LR classifiers. Integrating all extracted lexical and sentimental features resulted in the best performance across three experiments using varying input features (Text Length, Sentiment-Negative, Sentiment-Compound, and Words Percentage). Notably, the DT

method achieved a significant 79% accuracy. The study recommends broadening the approach by incorporating additional features like semantic and linguistic. It also highlights the imbalanced nature of the used dataset and proposes oversampling or undersampling to address this issue.

*Mitrović, Andreoletti & Ayoub (2023)* conducted two experiments to compare human-generated and ChatGPT-generated text using three datasets: 1,000 human-generated restaurant reviews from Kaggle, 350 ChatGPT-generated reviews from custom queries (ChatGPTquery), and 1,000 reviews where ChatGPT rephrased the human-generated reviews (ChatGPT rephrase). The study developed fine-tuned a Transformer-based ML model, DistilBERT, to classify short texts as either ChatGPT-generated (positive) or human-generated (negative). The model's performance was compared to a traditional perplexity-based classification approach, which measures how well a language model predicts text, with lower perplexity indicating better performance. The study found traditional perplexity-based methods ineffective and emphasized the superiority of ML models, specifically DistilBERT. ML-based classification showed much higher accuracy: 0.98 for customer queries *vs.* 0.84 for original human-generated texts, and 0.79 for rephrased texts *vs.* 0.69 for original human-generated texts. SHAP explanations revealed ChatGPT's polite and adaptable writing style across different domains, however noted its lack of nuanced human language features like irony and metaphors. A primary study limitation was its omission of exploration into alternative machine learning models beyond those utilized.

The study by *Gambetti & Han (2023)* aimed to assess a GPT-based text detector's ability to identify AI-generated fake restaurant reviews that mimic high-quality authentic ones. It used a dataset of 177,410 Yelp reviews from 5,959 restaurants, including 131,266 non-elite and 46,144 elite reviews. Using the OpenAI GPT-3 API, the study generated fake reviews by selecting 12,000 elite reviews randomly from the total of 46,144, using them as prompts to simulate genuine customer feedback. The study employed a fine-tuned GPTNeo model, an advanced variant of GPT-3 with 125 million parameters sourced from Huggingface (https://huggingface.co/EleutherAI/gpt-neo-125M). GPTNeo was trained to distinguish between fake and genuine restaurant reviews, leveraging its attention mechanism to selectively focus on key segments of input sequences, which is instrumental in NLP and computer vision tasks. The evaluation compared GPTNeo against traditional ML models like LR, NB, RF, XGB, and DL models such as BiLSTM, GPT-2, and RoBERTa to detect GPT-generated fake text (*Solaiman et al., 2019*). Traditional models used bag-of-words, while deep learning models used BPE for tokenization. GPTNeo models outperformed benchmarks, achieving 95.51% accuracy. The study compared AI-generated reviews with authentic ones across several dimensions: content, user characteristics, restaurant attributes, and writing style. AI-generated reviews tended to receive higher ratings, were authored by less established Yelp users, exhibited simpler language, and were more prevalent in low-traffic restaurants. The study acknowledges limitations arising from its exclusive focus on New York City may introduce regional biases and limit generalizability. Concerns also arise from the lack of transparency in the SafeGraph dataset's data collection methodology, impacting its representativeness. Using a single prompt template for

generating fake reviews and relying on elite reviews for GPT-3 input may have restricted content diversity. Additionally, the analysis period (2021–2022), influenced by the COVID-19 pandemic, may not fully capture typical consumer behaviors due to pandemic-related disruptions.

The recent work by *Buscaldi & Liyanage (2024)* explores the challenge of detecting fake reviews generated by LLMs, focusing on hotel reviews. The study utilized 400 positive and 400 negative reviews from platforms like TripAdvisor, Expedia, Hotels.com, Orbitz, Priceline, and Yelp, centered on 20 Chicago hotels. These reviews constituted the "truthful" subset of the Opinion Spam *corpus* created by *Ott et al. (2011)*. Fake reviews were generated using LLMs, including GPT-2, GPT-3, and TinyLLama, chosen for efficiency and low hardware requirements. Initial evaluation with a Multinomial Naïve Bayes model using TF-IDF achieved an F1 score of 96%, indicating effective differentiation based on vocabulary alone. Advanced transformer-based models (BERT, SciBERT, XLNet, and ELECTRA) were fine-tuned and tested, with average F1 scores ranging from 92.70 (XLNet) to 99.74 (BERT). Findings suggest that detecting fake reviews is feasible with sufficient training data, though GPT-2 reviews were challenging due to their similarity to human-written reviews. The study compared the log-probabilities of words in generated and non-generated classes, highlighting that generated reviews used more attributes (*e.g.*, "unhelpful", "terrible"), while non-generated reviews included more objects, places, and pronouns (*e.g.*, "door", "coffee", "she"). Using bi-grams and tri-grams, generated reviews often followed an "X was/were Y" pattern, where X typically denotes a service or hotel aspect and Y an adjective, contrasting with non-generated reviews' phrases like "in the room," revealing LLMs' tendency for recurrent patterns. The study concludes that generated reviews possess distinctive vocabulary and stylistic features, facilitating their detection with comprehensive training data. A limitation of the study is its narrow dataset, focusing solely on 20 Chicago hotels with 20 positive and 20 negative reviews each, potentially limiting generalizability to other locations and review types.

*Chen & Shu (2023)* also explored the complexities involved in detecting misinformation generated by LLMs. The study introduced the LLMFake dataset, which encompasses misinformation produced by seven types of LLMs: ChatGPT, Llama2-7b, Llama2-13b, Llama2-770b, Vicuna-7b, Vicuna-13b, and Vicuna-70b. This dataset includes instances generated through seven distinct methods: Hallucinated News Generation, Totally or Partially Arbitrary Generation, Paraphrase Generation, Rewriting Generation, Open-ended Generation, and Information Manipulation. Hallucination Generation (HG), which involves unintentionally nonfactual content due to LLM design and outdated information, often leading to inaccuracies in details like dates and names; Arbitrary Misinformation Generation (AMG), where malicious users intentionally create misinformation with or without specific constraints; and Controllable Misinformation Generation (CMG), techniques such as paraphrasing and rewriting used to preserve the original text's meaning while potentially obscuring authorship or enhancing deception. The misinformation generated by methods like Paraphrase Generation, Rewriting Generation, and Open-ended Generation draw upon data from well-known human-written misinformation datasets including Politifact, Gossipcop (*Shu et al., 2020*), and CoAID (*Cui & Lee, 2020*). Eight

**Table 1 Comparison of key studies on detecting fake reviews generated by LLMs.**

| Study | Dataset | Models used | | | | Key findings |
|-------|---------|----|----|----|----|------------|
| | | ML | DL | TF | EL | |
| *Bader et al. (2023)* | 4,000 English reviews of the Facebook mobile application from Google Play *vs.* 2,013 generated by ChatGPT | ✓ | | | | Decision tree achieved 79% accuracy |
| *Mitrović, Andreoletti & Ayoub (2023)* | 1,000 English reviews on restaurant from Kaggle *vs.* 395 ChatGPT-generated reviews, and 1,000 rephrased versions by ChatGPT | | ✓ | | | DistilBERT achieved accuracy 98% for ChatGPTquery dataset and 79% for ChatGPTrephrase dataset |
| *Gambetti & Han (2023)* | 24,000 reviews for New York restaurants, 12,000 elite Yelp reviews *vs.* 12,000 GPT-3 reviews | ✓ | ✓ | ✓ (RoBERTa, GPTNeo) | | GPTNeo achieved accuracy 95.51% |
| *Buscaldi & Liyanage (2024)* | 800 English reviews on Chicago hotels from platforms *vs.* 800 from GPT-2, GPT-3, TinyLLama, Paraphrased GPT-3 reviews | ✓ | ✓ | | | BERT base achieved highest F1 score of 99.74%; GPT-2 reviews hardest to detect |
| *Chen & Shu (2023)* | Approximately 731 of English pieces misinformation from politics and news dataset (*Shu et al., 2020*), and health dataset (*Cui & Lee, 2020*) *vs.* 3,031 pieces generated by ChatGPT, Llama2-7b, Llama2-13b, Llama2-70b, Vicuna-7b, Vicuna-13b, and Vicuna-33b | | | ✓ LLM (ChatGPT-3.5, GPT-4, Llama2-7B, Llama2-13B) | | Distinguishing LLM-generated misinformation from human-written content poses significant challenges |
| Proposed study | 20,000 Arabic reviews on Amazon products (*Alharthi, Siddiq & Alghamdi, 2022*) *vs.* 20,000 generated by ULMFit and GPT-2. | ✓ | ✓ | ✓ | ✓ | LR and CNN achieved an accuracy of 89.70%, akin to AraBERT's 90.0% |

representative LLM detectors were employed to evaluate the challenge of detecting both LLM-generated and human-written misinformation. These detectors include ChatGPT-3.5, GPT-4, Llama2-7B, and Llama2-13B. For each of these models, two strategies were considered: "No CoT" (no contextual understanding of the text) and "CoT" (with contextual understanding techniques applied) with zero-shot prompting strategies. The study employed the Success Rate % metric to gauge the likelihood of correctly identifying LLM-generated or human-written misinformation. The results showed that the detectors face significant challenges in detecting LLM-generated misinformation across various generation methods, particularly with Hallucinated News Generation, Totally Arbitrary Generation, and Open-ended Generation. For instance, ChatGPT-3.5 (or GPT-4) detects only 0.0% (or 10.0%) of hallucinated news, highlighting the difficulty in recognizing subtle hallucinations with LLM detectors. However, the study's exclusive focus on LLMs as detectors neglects the potential advantages of traditional, deep learning, or ensemble methods. These alternative approaches may offer better generalization across diverse types of misinformation, which LLMs could struggle to achieve due to their specific training and generation characteristics.

The related work on detecting fake reviews is summarised in Table 1, which highlights a trend towards employing traditional ML, DL, and transformer models like DistilBERT, GPTNeo, and BERT to address the prevalence of AI-generated deceptive content. These studies have demonstrated high accuracies in distinguishing between human-written and

AI-generated reviews across diverse domains using AI. However, there is a significant gap in exploring ensemble techniques such as soft voting, which could potentially enhance model performance by aggregating predictions from multiple models. This underexplored area presents an opportunity to bolster the robustness and reliability of fake review detection systems. Moreover, addressing the limitations of dataset domain diversity and imbalance is crucial for improving the generalizability and effectiveness of these models. This necessitates exploring datasets across various review domains and languages, including those less commonly studied, such as Arabic.

# METHODOLOGY

## Mathematical formulation of the proposed algorithm

To classify human and computer-generated reviews using an ensemble learning technique, specifically voting, the implemented problem statement can be mathematically expressed as follows:

Let $D = \{(x_1, y_1), (x_2, y_2), \ldots, (x_n, y_n)\}$ be a dataset of reviews, where $x_i$ represents the $i$-th review text and $y_i$ represents its corresponding label (*e.g.*, 0 for computer reviews and 1 for human reviews). Each review $x_i$ is represented as a feature vector $x_i = (w_{i1}, w_{i2}, \ldots, w_{im})$, where $m$ is the number of features (words or tokens) in the review.

The goal of classifying human and computer-generated reviews is to learn a model $f : X \rightarrow Y$, where $X$ is the space of feature vectors and $Y$ is the set of class labels (0 for computer and 1 for human reviews). The model $f$ assigns a class label $\hat{y}$ to a new review $x$ based on the votes of multiple base classifiers. In this study, the base classifiers are ML or DL classifiers.

Let $C = \{c_1, c_2, \ldots, c_k\}$ be a set of $k$ base classifiers. Each base classifier $c_i$ is a function $c_i : X \rightarrow Y$ that predicts a class label for a given feature vector $x$.

The voting method combines the predictions of the base classifiers to make a final prediction $\hat{y}$ for a new review $x$. This can be expressed as Eq. (1):

$$\hat{y} = \text{argmax}_{y \in Y} \sum_{i=1}^{k} \delta\left(c_i\left(x\right), y\right) \tag{1}$$

where $\delta\left(c_i\left(x\right), y\right)$ is a signal function that is 1 if the prediction of base classifier $c_i$ for review $x$ is equal to class label $y$, and 0 otherwise. The final prediction $\hat{y}$ is the class label with the highest sum of votes across all base classifiers.

This study's objective is to compile both ML, DL, and transformer models, as well as EL techniques, to classify Arabic textual reviews as either human-generated or computer-generated. Additionally, hybrid EL is applied to aggregate predictions with both ML and DL models. Employing this ensemble technique aims to access whether it can improve the accuracy of detecting computer-generated reviews. The architecture outlined in Fig. 1 illustrates a systematic process for the study's methodology, emphasizing the key processes and techniques utilized, which are further explained in the subsequent sections.

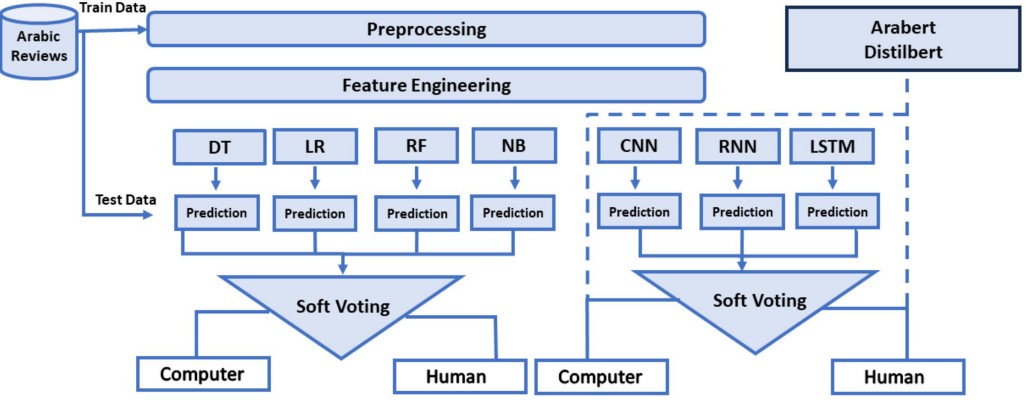

**Figure 1 Architecture of the human and computer-generated reviews classification models.**

## Dataset

The dataset used in this study was provided by the authors of *Alharthi, Siddiq & Alghamdi (2022)* upon request. They translated the publicly available 2018 Amazon Review Data (*Ni, Li & McAuley, 2019*) into Arabic using the Google Translate API (https://cloud.google.com/dotnet/docs/reference/Google.Cloud.Translation.V2/latest) for their research on detecting fake Arabic reviews. The dataset consists of approximately 40,000 reviews. The uniqueness of this dataset lies in the labeling of reviews as "original" reviews posted by humans and "fake" reviews generated by two language models, ULMFit (*Howard & Ruder, 2018*), which is a technique for fine-tuning a pre-trained language model to perform a particular downstream task, and GPT-2 (*Antoun, Baly & Hajj, 2020*), a specific type of large language model developed by OpenAI. The dataset covers reviews on various product categories, including Kindle Store Books, Pet Supplies, Home and Kitchen, Electronics, Sports and Outdoors, Tools and Home Improvement, Clothing Shoes and Jewellery, Toys and Games, and Movies and TV. The metadata of the provided dataset includes category, rating (1 to 5), class, and reviews in text format.

Table 2 presents a description of the dataset, including the number of reviews labeled as "Human" and "Computer". The dataset consists of average unique words, average sentences per review, and a sample review in Arabic and English for each class.

## Data preprocessing

Initially, we focused solely on reviewing textual content and labels, omitting category and rating data. Since reviews often include noisy elements like emojis and punctuation marks, employing preprocessing techniques was crucial before using them in AI models. *Vijayarani, Ilamathi & Nithya (2015)* suggest text preprocessing techniques to organize noisy data while preserving its original meaning. Their proposed process converts unstructured text into structured data, enhancing data quality for easier pattern recognition and information extraction. Three common preprocessing techniques for Arabic text, including tokenization, normalization, and removal of unnecessary words,

**Table 2 Dataset overview for human and computer-generated reviews.**

| | Human reviews | Computer reviews |
|---|---|---|
| No. of reviews | 20,224 | 20,216 |
| Average no. of unique words | 9 | 9 |
| Average no. of sentences | 2 | 2 |
| Review sample in Arabic | يعمل بشكل رائع مع جميع شموع البطاريات الخاصة بي من نفس العلامة التجارية | هذه الوسادة أنقذت ظهري. أنا أحب شكل وملمس هذه الوسادة |
| Review sample in English | Works great with all the candle batteries from the same trademark | This pillow saved my back. I love the shape and texture of this pillow. |

were applied to the reviews using the Tasaheel tool (*Himdi & Assiri, 2023*). A brief description of each technique is provided below:

1. **Tokenization:** It involves breaking text into smaller units called tokens.

2. **Normalization:** In Arabic text, normalization aims to standardize the data for uniformity and consistency. It typically involves several steps, such as removing diacritics, which are marks representing vowels and phonetic information in Arabic words. Simplifying text processing by eliminating diacritics is a common practice. Additionally, normalization ensures that Arabic letters are represented in a standardized form, as these letters can have multiple forms depending on their position within a word. For example, normalization helps maintain a consistent representation of the Arabic letter alef ((أ)), which can appear in different forms depending on its position in a word.

3. **Removel of numerical data, non-alphabetic characters, and stop words:** The removal of numerical data results in enhancing the quality of datasets, as highlighted by *Sudheesh et al. (2023)*. Therefore, all numerical data were eliminated from the dataset text, as they do not contribute meaningful information to the decision-making process. Furthermore, to improve dataset quality and model performance, non-alphabetic characters such as punctuation marks, special characters [?, @, #, /, &, %], and Uniform Resource Locators (URLs) were removed. This preprocessing step aims to streamline the dataset by eliminating irrelevant elements. Additionally, Arabic stop words were excluded during preprocessing, aligning with findings by *Sudheesh et al. (2023)*. Eliminating stop words not only enhances model accuracy and training efficiency by retaining only relevant information but also allows for more in-depth analysis, particularly beneficial for a limited dataset (*Kadhim, 2018*).

## Feature engineering

It is important to note that the feature extraction approaches used for ML and DL differed. For ML, text features are extracted from the cleaned reviews using the Term

Frequency-Inverse Document Frequency (TF-IDF) approach (https://scikit-learn.org/stable/modules/generated/sklearn.feature_extraction.text.TfidfTransformer.html#sklearn.feature_extraction.text.TfidfTransformer). TF-IDF is employed to capture valuable features from the data, thereby enhancing the performance of AI models. Furthermore, TF-IDF assigns weights to words based on their relative importance within a document and across the entire *corpus* (*Wu et al., 2008*). Notably, this technique aims to highlight words that are more distinctive and informative, while moderating common words that carry less meaning.

Mathematically, TF-IDF for a term $t$ in a document d is calculated using Eq. (2):

$$\text{TF} - \text{IDF}\,(t,\,d) = \text{TF}\,(t,\,d) \times \text{IDF}\,(t) \tag{2}$$

where, $\text{TF}(t,\,d)$ is the term frequency of $t$ in $d$, representing the raw count of t's occurrences in $d$.

$\text{IDF}(t)$ is the inverse document frequency of $t$, calculated using Eq. (3):

$$\text{IDF}\,(t) = \log\left(\frac{N}{\text{df}(t)}\right) \tag{3}$$

where:

$N$ is the total number of documents in the *corpus*, and $\text{df}(t)$ is the number of documents containing $t$.

This approach not only reduces processing time and effort but also aids in improving the effectiveness of the models (*Karamibekr & Ghorbani, 2012*).

On the other hand, the process of preparing textual data for DL models involved a tokenizer, padding sequences, and converting the labels to a categorical format using Keras (https://keras.io/). The selection of these feature engineering techniques was based on their frequent use in the field of text classification and their proven effectiveness in many domains (*Shaukat et al., 2020*; *Mohawesh et al., 2023*).

## Model compilation

The aforementioned previous works revealed that detecting human and computer-generated text employs prevalent methods and uses classification techniques that rely on characteristics and binary classification since this includes categorizing the reviews as either human or computer-generated. The ML, DL, and TF classifiers employed in this study possess the capability to effectively perform good in this task due to their capacity to efficiently process and examine substantial volumes of data, enabling them to find discernible patterns and distinctive characteristics that may differentiate human generated reviews from computer-generated ones. Several models were compiled to distinguish between human and computer-generated reviews by utilizing ML, DL, TF, and EL techniques described in the following section:

### Machine learning

A brief description of each ML model is provided below:

- **Naive Bayes:** It is composed of a number of algorithms that are based on the Bayes Theorem and assume independence of the attributes. It is based on estimates, in which

**Table 3 Hyperparameters for ML classifiers.**

| Model | Hyperparameter tuning |
| --- | --- |
| LR | C': 10, 'penalty': 'l2', solver: 'liblinear', random_state: 50 |
| DT | criterion': 'entropy', 'max_depth': 10, 'min_samples_leaf': 2, 'min_samples_split': 5, 'random_state': 100 |
| RF | max_depth': 15, 'n_estimators': 125, 'random_state': 50 |
| NB | alpha: 1.0 |

the model adjusts its probability table using the training data and predicts new observations by estimating the class probability based on its feature values. The small amount of training data required by NB leads to significant storage space savings, faster results, and robustness to missing data (*Osisanwo et al., 2017*).

- **Logistic regression:** It is a classifier that establishes a link between features and likelihood of the outcome (*Pramanik et al., 2021*). It is based on the logistic function, an S-shaped curve that maps real value numbers to values between 0 and 1 (*Machine Learning Plus, 2021*). LR is reliable for classifying problems (*Grover, 2020*), and it prevents over-fitting.

- **Decision trees:** It is a supervised learning algorithm for classification and regression tasks (*Swain & Hauska, 1977*). It adeptly discerns various ways of dividing datasets under changing conditions. Furthermore, it strategically selects the best attribute, placing it at the tree's root, and subsequently partitions the training into subsets based on dataset feature values. The impact of nodes within the tree becomes more pronounced when they are closely positioned.

- **Random Forest:** It utilizes a decision tree model based on bootstrap aggregation techniques, also known as bagging (*Genuer et al., 2017*). This ensemble method combines predictions from multiple decision trees, reducing high variance and enhancing robustness against outliers.

These ML have been employed due to their proven effectiveness in similar tasks (*Bader et al., 2023*; *Mohawesh et al., 2023*). GridSearchCV from Skilleran (https://scikit-learn.org/stable/modules/generated/sklearn.model_selection.GridSearchCV.html) was used to optimize the predictive capabilities of these models by shortlisting the optimal hyperparameters. This technique explores various hyperparameter combinations, identifying those that yield superior predictive accuracy for each model. The outcomes of this hyperparameter tuning are detailed in Table 3.

### Deep learning

A description of each classifier is detailed:

1. **Recurrent neural networks:** It utilizes its internal memory to handle inputs of varying lengths effectively for classification purposes. They are considered Turing complete,

| Table 4 Hyperparameters for DL classifiers. | | |
|---|---|---|
| Model | Trainable parameters | |
| LSTM | 74,602 | batch_size = 64, epochs = 6, loss = 'categorical_crossentropy', optimizer = 'adam', Embedding = 500, 120. Learning rate = 0.001 activation function = ReLU |
| RNN | 63,152 | |
| CNN | 72,245 | |

meaning they can simulate any arbitrary program. It can also handle random input sequences by running these programs (*Pascanu et al., 2013*).

2. **Long short-term memory:** It is a type of RNN that captures sequential relationships among words within a sentence (*Hochreiter & Schmidhuber, 1997*). Given that textual information can be treated as time-series data, the order of words holds a pivotal role in shaping the meaning of sentences. The LSTM cell comprises four crucial components: the forget gate, output gate, input gate, and update gate. The forget gate determines what to discard from the previous memory units, the input gate decides what information to incorporate into the neuron, the update gate modifies the cell, and the output gate produces the new long-term memory.

3. **Convolutional neural networks:** It is structured with a series of interconnected layers, where the output from one layer serves as input for the next. These layers include a convolutional layer, a pooling layer, and a fully connected layer, as outlined in the study. The initial convolution layer incorporates multiple filters, 16 for different regions, each with two filters, to extract sentence features. These filters convolve the input, generating feature maps of variable lengths. The subsequent layer, max pooling, captures essential features from the prior maps. The last layer is a dense (fully connected) layer utilizing the sigmoid function as an activation function (*Han & Moraga, 1995*). This layer produces the network's output, indicating whether the input sentence is positive or negative (*Krizhevsky, Sutskever & Hinton, 2012*; *Mohammed & Kora, 2019*).

These DL models exceeded in classifying machine-generated text in various studies (*Mitrović, Andreoletti & Ayoub, 2023*; *Mohawesh et al., 2023*), specifically long short-term memory (LSTM) networks, recurrent neural networks (RNNs), and CNNs. Table 4 provides a comprehensive breakdown of the parameters associated with all the DL models employed in this study. The dataset was then imported into the model layer with an output size of 512. The output was then inputted into the dropout layer with a dropout rate of 0.2. The dropout layer was used to mitigate the issue of model overfitting. Subsequently, the output was sent to an additional model layer, which had an output size of 256. It was then passed to another dropout layer with a dropout rate of 0.2. Subsequently, the data was sent to a dense layer using the rectified linear unit (ReLU) activation function in order to diminish the output to 64. The next phase was transmitting the output to the dense layer *via* the softmax algorithm.

### Transformers based models

1. **Arabert:** BERT is a Bidirectional Encoder Representation from Transformers (*Devlin et al., 2019*). The bi-directional relation allows the Pre-Trained Embedding to train on the provided text's left and right context to better understand its variation. Furthermore, it continues learning by applying unsupervised learning on unlabeled text, allowing it to improve its performance and making it suitable for applications like Google Search. The model employed in this study is the Arabertv2-Base from 'aubmindlab/bert-base-arabertv2' (https://huggingface.co/aubmindlab/bert-base-arabertv2) pre-trained transformer model. This model comprises 12 layers (transformer blocks), each with 12 attention heads. It has a hidden size of 768, totaling approximately 110 million parameters. Trained on a comprehensive and varied dataset of 77 GB of Arabic text, which includes Arabic Wikipedia, news articles, and various web pages, spanning a wide range of topics and styles in Modern Standard Arabic (MSA). The model uses a WordPiece tokenizer optimized for Arabic. Its pre-training follows the standard BERT objectives: Masked Language Modeling (MLM) and Next Sentence Prediction (NSP). MLM involves randomly masking some tokens in the input and training the model to predict these masked tokens, while NSP involves training the model to determine if a pair of sentences are consecutive in the original text. Notably, it incorporates the Farasa Segmenter and Farasa POS Tagger (*Darwish et al., 2017*), which enhance the handling of Arabic-specific text characteristics, thereby improving tokenization and overall model performance.

2. **Distilbert:** Is a lightweight variant of BERT that uses minimal resources to generate comparable results to the BERT model (*Sanh et al., 2019*). It uses the distillation method, creating small models to mimic the process and representation of the larger BERT model. Each smaller model learns from BERT and updates its weight using the following parameters: Distillation loss, similarity loss, and masked language model mask. The lightweight nature of Distilbert makes it suitable for resource-constrained devices. In this study, the DistilBERT-base-uncased (https://huggingface.co/distilbert/distilbert-base-uncased) was utilized, as it has a reduced number of layers approximately six transformer layers, each incorporating 12 attention heads, with a hidden size of 768. It was trained using MLM and additional distillation techniques to align its capabilities with the larger BERT model. The pre-training process for DistilBERT involves a diverse range of texts, including news articles, social media content, and web pages. This comprehensive dataset ensures a thorough understanding of the nuances of the language. Both pre-trained transformer-based parameters were set as batch size = 16, maximum length = 128, and number of epochs = 3.

### Ensemble learning

The EL technique employed in this study uses a soft voting approach, which involves two combinations of models. The first combination utilized ML models (RF, DT, LR, and NB), while the second combination employed DL models (CNN, LSTM, and RNN). The best-performing classifiers, determined by their accuracy scores, were chosen as the estimators for the voting model. Figure 2 demonstrates the development process of the

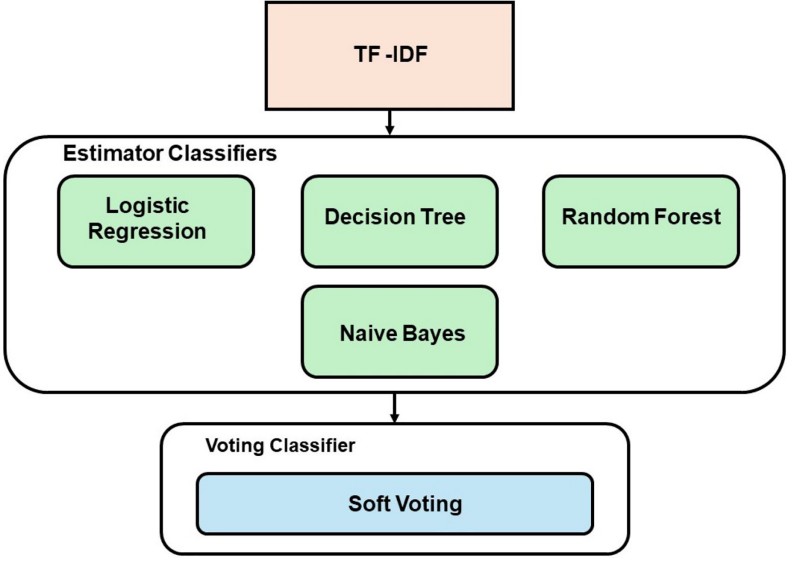

**Figure 2  Voting model development with ML.**     

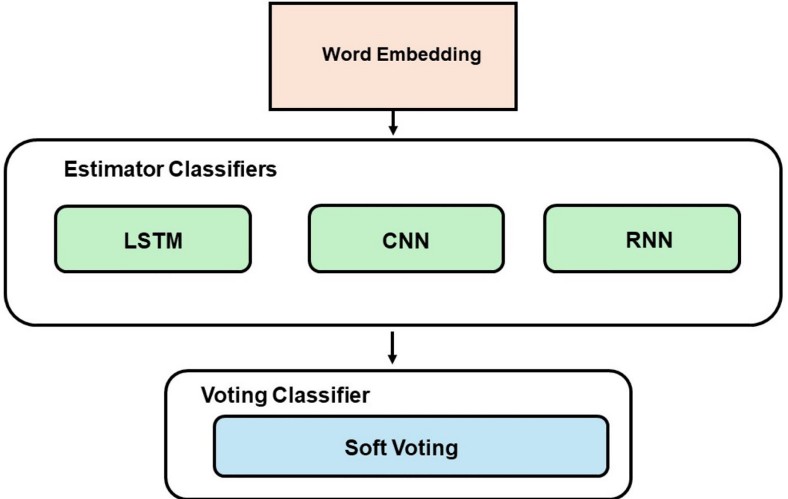

**Figure 3  Voting model development with DL.**     

voting classifier using ML estimators, while Fig. 3 depicts the same technique using DL estimators. Furthermore, a hybrid ensemble learning approach is employed to combine predictions from both the ML and DL models that demonstrate the best performance.

Stronger generalization competence may be obtained by merging many classifiers using ensemble techniques (*Dietterich, 2000*), as opposed to aiming to identify the single optimal classifier for a given classification problem. Ensemble approaches often provide more precise outcomes compared to using a single classifier. The models compiled through this learning method applied the voting technique to assess whether this strategy can improve the accuracy of ML and DL models. Two frequently used voting methods are hard voting and soft voting (*Żabiński et al., 2020*). The voting ensemble technique (*Manconi et al.,*

*2022*; *Kumari, Kumar & Mittal, 2021*) is the most endorsed method for combining models, since it offers models with a high capacity for generalization. It combines many estimators to generate predictions, by permitting the models to vote for the predicted class (*Kabari & Onwuka, 2019*). Hard voting implies the final classification prediction is chosen by a majority vote from many estimator models. Its equation is presented in Eq. (4):

$$\widehat{y}_i = \{C_1(x_i),\ C_2(x_i),\ \ldots,\ C_j(x_i),\ \ldots,\ C_m(x_i)\} \tag{4}$$

where $m$ is the number of classifiers, $x_i$ denotes the $i^{th}$ sample, $C_j(x_i)$ denotes the predictions of the $j^{th}$ classifier, and the mode function is used to calculate the majority vote of all prediction (*Yu et al., 2020*).

Soft voting involves computing the weighted total of the prediction probabilities from all classifiers for each class in order to estimate the final class label, as seen in Eq. (5). The label associated with the class with the greatest overall probability is chosen.

$$\widehat{y}_i = \arg\max_k \sum_{j=1}^{m} w_j p_{i,k}^j \tag{5}$$

where $m$ is the number of classifiers, $w_j$ denotes the weight of $j^{th}$ classifier, $p_{i,k}^j$ donates the prediction probability of $j^{th}$ classifier for assigning the $i^{th}$ sample to $k^{th}$ class (*Yu et al., 2020*).

We have selected soft-voting EL as our EL approach because of its exceptional performance in text classification tasks (*Bountakas & Xenakis, 2023*). It computes the average output probabilities generated by all the base learners presented in the ML and DL classifiers. Furthermore, it reduces the structural sensitivity caused by the base classifier and decrease the variance of the ensemble developed by multiple employed classifiers (*Das et al., 2019*). Hence, it is especially effective when handling a small cohort of well-calibrated classifiers with scores that correlate with probability estimations and produce more reliable and efficient results by taking each classifier's confidence levels into account when generating predictions.

All models were compiled using a server endowed with an Intel(R) Xeon(R) E5-2670 CPU, NVIDIA(R) GeForce GTX 1080 GPU with 8 GB video memory, and 64 GB RAM. The models were utilized by Google Colab (https://colab.google/) running on Python 3.0.

## Model evaluation

The organized dataset included 20 k human and 20 k computer Arabic-generated reviews. To test and evaluate each compiled model, the dataset was split into 80% (32 k reviews) for training, 10% (4 k reviews) for validation, and 10% (4 k reviews) for testing. To assess the models' performance in all experiments, we employed standard metrics such as accuracy, precision, recall, and F1 scores, presented in Formulas (6)–(8), respectively. These metrics were calculated by comparing expected and measured results, enabling the analysis of prediction accuracy within the training sample. The classification into four groups—true positive (TP), true negative (TN), false positive (FP), and false negative (FN) —facilitated the derivation of these measures.

1. Accuracy: measures a model's overall performance, calculated as the ratio of correctly predicted instances (true positives and negatives) to the total instances. It is calculated using Eq. (6)

$$\text{Accuracy} = \frac{TP + TN}{TP + TN + FP + FN} \tag{6}$$

2. Precision: focuses on the positive predictions made by the model. It is calculated by taking the ratio of correctly predicted positive instances among all instances predicted as positive, as given in Eq. (7).

$$\text{Precision} = \frac{TP}{TP + FP} \tag{7}$$

3. Recall: indicates the proportion of correctly predicted positive instances among all actual positive instances, as given in Eq. (8).

$$\text{Recall} = \frac{TP}{TP + FN} \tag{8}$$

## TEXTUAL ANALYSIS

Based on the previous research in the field of linguistic fingerprinting, it is assumed that each individual has a distinct way of using language to convey their thoughts, opinions, and ideas, which means that the linguistic methods employed to communicate vary among humans. This implies that linguistic methods of communication vary among humans. Furthermore, analyzing linguistic fingerprints through textual analysis enhances performance in classification tasks, such as recognizing certain individuals (*Yang, Dragut & Mukherjee, 2020*) and predicting their opinions (*Rocca & Yarkoni, 2022*). Hence, we conduct an analysis of specific linguistic features to distinguish between human-written and computer-generated content, following the methodology employed by *Himdi et al. (2022)*. The previous study employed specific linguistic features to train models that detect Arabic fake news. The three word-use linguistic categories used to implement textual analysis are described below:

- **Part of speech (POS):** According to a study by *Alsmearat et al. (2017)*, the frequency distribution of POS identifiers varies based on genre to detect author attributions, reflecting the composition of sentences with verbs, nouns, adjectives, adverbs, prepositions, conjunctions, and pronouns. In this study, we examine potential correlations between human and machine-generated reviews by constructing features for each post based on the frequency of each POS tag, serving as a baseline for evaluating automated methods.
- **Emotions:** Emotional features assess the level of emotion conveyed in a text, examining six fundamental human emotions-anger, disgust, fear, sadness, joy, and surprise-to evaluate the emotional tone of each post.

- **Linguistics:** Linguistic features reveal the semantic meaning of sentences by identifying relationships between concepts, encompassing hedges, assurances, temporal and spatial words, exceptions, negations, illustrations, intensifiers, oppositions, justifications, and superlatives.

We conduct two approaches to examine word use between humans and computers, which are lexical diversity and lexical density. Lexical Density ($L$) is the ratio of lexical words, also called lexical items, divided by the total number of words in the text (*Johansson, 2008*), as given in Eq. (9).

On the other hand, lexical diversity is a measure of the number of unique words used in a text against the total number of words. Specifically, it measures the lexical richness of using a variety of words in a text (*Johansson, 2008*). In an effort to calculate lexical diversity, Hypergeometric Distribution Derived Distribution (HD-D) was calculated by *McCarthy & Jarvis (2010)*. It is derived from the hypergeometric distribution and offers a more resilient metric for evaluating lexical diversity that stays reasonably consistent regardless of the length of the text. Due to the presence of reviews with different lengths in this dataset, we opted for the HD-D assessment. Both textual measures may be mutually beneficial, as they may be advantageous when applying them to carry out an in-depth textual analysis of the dataset. The lexical density and lexical diversity of word categories in both classes within the dataset are detailed in Eqs. (9) and (10), respectively:

$$\text{Lexical density} = \left( \frac{\text{Total number of occurrences of each feature in a class}}{\text{Total number of words in the whole class}} \right) \times 100 \qquad (9)$$

$$HD - D = \frac{1}{T} \sum_{i=1}^{T} \left( 1 - \frac{\binom{N - k_i}{n}}{\binom{N}{n}} \right) \qquad (10)$$

where:

- $T$ is the total number of unique word types.
- $k_i$ is the frequency of the $i$-th word type.
- $N$ is the total number of words in the text.
- $n$ is the sample size.

## RESULTS

This section presents the outcomes of the conducted experiments using different approaches (ML models, DL models, TF-based models, and ensemble and hybrid approaches for the previous models) to evaluate their performance in classifying human and ML-generated reviews. Additionally, it presents a comprehensive textual analysis of the human and computer-generated reviews.

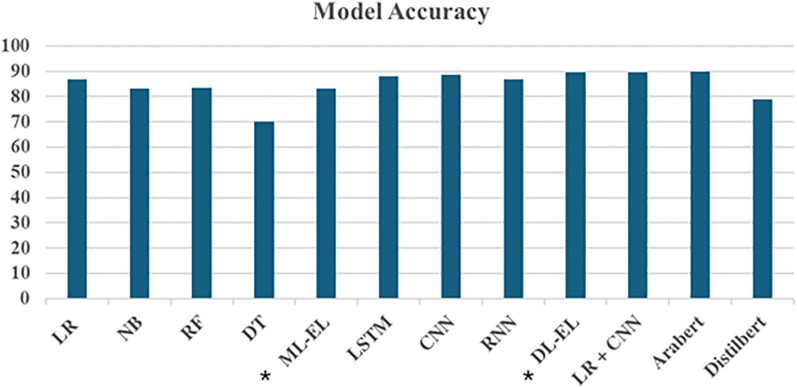

**Figure 4 Accuracy results of models.** *ML-EL, machine learning ensemble learner; DL-EL, deep learning ensemble learner.                

## Models classification results

Figure 4 demonstrates the accuracy of various ML, DL, and TF models, including ensemble and hybrid methods, using TF-IDF and embedded features. The x-axis represents different models, while the y-axis indicates their corresponding accuracies.

The figure shows that LR exhibited the highest accuracy of 86.85% among all the ML models utilizing TF-IDF features. RF and NB followed closely with accuracies of 83.32% and 83.12%, respectively. In contrast, DT had the lowest performance, with an accuracy of 70.08%. The ensemble model, which combines RF, DT, LR, and NB, achieved an accuracy of 83.19%. Notably, LR's accuracy surpasses the ensemble method's, suggesting that LR is particularly robust for classifying Arabic textual reviews. A possible reason for for LR's superior performance is its effectiveness in high-dimensional spaces, which aligns well with TF-IDF features that often result in sparse and high-dimensional data.

DL models employing embedded features also show promising results. Both LSTM and CNN emerged as top performers, attaining accuracy rates of 88.15% and 88.55%, respectively. The RNN achieved a commendable accuracy of 86.88%, slightly trailing behind LSTM and CNN. Remarkably, the EL, synthesizing predictions from LSTM, CNN, and RNN, demonstrated superior performance compared to individual DL models, boasting an accuracy of 89.57%. This highlights the effectiveness of combining multiple DL architectures to capture diverse patterns in the data.

On the other hand, the hybrid model, which integrates LR and CNN, leverages the strengths of both ML and DL approaches. This hybrid model achieved an accuracy of 89.7%, demonstrating improved predictive performance through synergy and robustness in handling different aspects of model performance. This method rivals advanced TF models like AraBERT and DistilBERT. AraBERT achieved the highest accuracy overall at 90.0%, while DistilBERT, though lower, still maintained a significant presence with an accuracy of 79.0%. In addition to the accuracy, Table 5 further evaluates the above-mentioned models based on their precision and recall. This demonstrates the superiority of our EL models, which are comparable to transformer-based models.

**Table 5 Models evaluation.**

| Model | Accuracy (%) | Precision (%) | Recall (%) |
|---|---|---|---|
| Machine learning | | | |
| LR | 86.85 | 84.15 | 87.34 |
| NB | 83.12 | 84.76 | 82.44 |
| RF | 83.32 | 83.31 | 84.33 |
| DT | 70.08 | 71.74 | 69.99 |
| EL | 83.19 | 85.74 | 80.27 |
| Deep learning | | | |
| LSTM | 88.15 | 90.81 | 88.10 |
| CNN | 88.55 | 89.74 | 88.61 |
| RNN | 86.88 | 85.75 | 78.91 |
| EL | 89.57 | 90.76 | 88.94 |
| Hybrid | | | |
| LR + CNN | 89.70 | 88.3 | 91.4 |
| TF | | | |
| AraBERT | 90.0 | 94.0 | 87.0 |
| DistileBRT | 79.0 | 77.0 | 81.1 |

**Table 6 Comparison of word use categories between human and computer reviews.**

| Word categories | Human reviews | Computer reviews |
|---|---|---|
| **POS** | | |
| Nouns | 10.46 | 13.99 |
| Proper noun | 8.6 | 10.16 |
| Verbs | 3.4 | 6.9 |
| Conjunction | 5.8 | 2.5 |
| Pronoun (all) | 3.16 | 4.3 |
| Pronouns (Singular) | 2.44 | 4.7 |
| Pronouns (Plural) | 2.9 | 1.07 |
| Adverb | 7.03 | 2.2 |
| Adjectives | 0.46 | 6.3 |
| **Linguistics** | | |
| Assurance | 0.01 | 0.03 |
| Negators | 1.14 | 0.07 |
| Opposite | 0.15 | 0.62 |
| Justification | 0.25 | 0.03 |
| Exception | 0.11 | 0.12 |
| Illustration | 0.18 | 0.19 |
| Hedge | 2.5 | 0.05 |
| Order | 0.01 | 1.2 |
| Intensifiers | 6.8 | 0.01 |

(Continued)

| Word categories | Human reviews | Computer reviews |
|---|---|---|
| Quantity | 0.07 | 5.06 |
| Religious names | 0.7 | 0.032 |
| Location | 1.8 | 0.009 |
| Month | 1.3 | 0.07 |
| Day | 0.8 | 0.02 |
| **Emotions** | | |
| Joy/happiness | 1.16 | 0.03 |
| Sad/sadness | 6.4 | 1.9 |
| Anger | 3.7 | 2.7 |
| Fear | 3.0 | 1.7 |
| Surprise | 2.5 | 2.2 |
| Disgust | 0.1 | 4.9 |

## Data textual analysis results

Textual analysis was conducted to compare human-generated and machine-generated reviews. Table 6 displays the lexical densities across POS tags, linguistic features, and emotional expressions in both types of reviews.

The analysis of linguistic and emotional features in the dataset reveals key disparities between human and computer-generated textual content. Notably, human-generated reviews express lower lexical densities across various linguistic categories than computer-generated reviews, such as nouns (10.46% *vs.* 13.99%), proper nouns (8.6% *vs.* 10.16%), verbs (3.4% *vs.* 6.9%), and all types of pronouns (3.16% *vs.* 4.3%). This indicates that humans tend to write reviews focusing on a specific aspect of the product, resulting in fewer lexical components. Moreover, key findings include the prevalence of emotional words such as sadness (6.4% *vs.* 1.9%), anger (3.7% *vs.* 2.7%), joy (1.16% *vs.* 0.03%), and surprise (2.5% *vs.* 2.2%) in human reviews, which show higher emotions compared to computer-generated reviews. These findings collectively underscore the distinct linguistic and emotional patterns inherent in human and computer-generated textual data, providing valuable insights for discerning between human and computer-generated content.

A deeper review of the textual analysis results raises several findings in terms of the difference between human and computer language. First, we find that computer-generated reviews are attentively structured according to the typical review structure. The reviews adhere to a typical review format, consisting of either all positive or all negative feedback. While the general layout of the reviews remains uniform, it is noteworthy that computer reviews produce more rigorous reviews than human-written reviews, which exhibit a less rigid style in conveying reviews. The insights discovered indicate that the use of discourse markers, such as conjunctions, intensifiers, hedges, and justification terms, corresponds closely with a decrease in the logical coherence seen in human reviews. However, we find an increased use of quantifiers, verbs, and adjectives that were commonly used to create the

formal style of computer reviews. The adjectives were to characterize the product, verbs to illustrate their use and quantity to represent some descriptions of the product, all of which are used to imitate the human-generated reviews. We anticipate that the reason for this occurrence might be attributed to the rigorous structure of the computer-generated reviews. It, therefore, provides explicit, clear, and less creative content, minimizing the need for discourse markers often used in explicable and more creative reviews written by humans. In fact, human-generated reviews contained more details of location, month, and day, creating more specifics in reviews compared to computer reviews. This notion aligns with the study of *Herbold et al. (2023)*, which found that computer-generated essays were composed of a more strict language use compared to human-generated essays. It also aligns with the study of *Liao et al. (2023)* that found that medical texts written by humans were more diverse than the ones produced by computers. In general, the human expressive tone is more evident in their reviews, as opposed to the formal tone prevalent in computer reviews.

Second, terms related to society and details of events such as time, location, month, and religious names are profoundly less common in computer reviews compared to humans. The limitation of computers is their incapacity to fully comprehend contextual information, which hinders their ability to provide coherent responses. Computers are limited to generating content only based on the data they have been trained on and are unable to possess the capacity to fully understand the intricate details of language and culture, in contrast to people who use these notions to provide nuanced clarifications in their reviews. Consequently, computers have the potential to generate text that adheres to grammatical rules but lacks cultural adaptability. The aforementioned findings are consistent with the insights revealed in an article that introduced the limitations in AI-generated content (https://www.linkedin.com/pulse/challenges-limitations-ai-generated-content-pradeep-kumar/).

Third, a noteworthy observation was the use of hedges. The findings of the textual analysis indicate that human reviews had a higher percentage of hedges, specifically 2.5%, compared to 0.05% in computer reviews. One plausible explanation is that individuals may be aware of guilt when writing negative feedback that could negatively impact the company's reputation. As a result, they may use more cautious language (hedges) to distance themselves from this guilt as well as offer an incentive to rescind their reviews (https://www.nationalstrategic.com/why-would-they-write-that-the-psychology-of-customer-reviews/).

Finally, we find an increased use of singular pronouns compared to plural pronouns (4.7% *vs.* 1.07%) in computer reviews. A possible explanation might be that computer-generated reviews employ first-person pronouns as a linguistic standard in order to improve the communicative and accessible nature of the experience for users (https://wac.umn.edu/teaching-writing/teaching-resources/what-chatgpt-and-how-does-it-work). It facilitates the establishment of a feeling of engagement and connection between them and human users, which can be reflected in their generated reviews.

In terms of lexical diversity, we calculated the HD-D and found that computer-generated reviews scored 0.212, whereas human-generated reviews scored 0.645. The score

of 0.212 for computer-generated reviews indicates low lexical diversity, characterized by a mix of repeated and unique terms but lacking particularly rich and varied wording. Conversely, the score of 0.645 for human-generated reviews suggests a high level of lexical diversity, indicating a wide range of vocabulary and sophisticated use of language.

These findings support the earlier insights that human reviews tend to convey emotions and personal experiences with products, leading to greater lexical diversity compared to computer-generated text. Despite human-generated reviews having lower lexical density than computer-generated ones, they exhibit greater lexical diversity. This suggests that humans tend to focus their reviews on specific aspects of products, incorporating personal perspectives, emotions, and needs.

In contrast, computer-generated reviews are formulated strategically without focusing on specific product features, resulting in a higher word count but less lexical diversity compared to human reviews.

## DISCUSSION

This research contributes to the identification of language patterns in online Arabic reviews generated by computers and humans, aligning with the approach adopted by *Himdi et al. (2022)* to analyze linguistic differences between human and computer-generated texts. Notably, our analysis reveals significant disparities in lexical and emotional features between human and computer-generated textual content. For example, computer-generated texts exhibit higher lexical densities across various linguistic categories, such as nouns, verbs, and pronouns, suggesting more word usage compared to human-generated texts. Additionally, our findings highlight differing emotional expressions, with human texts demonstrating higher emotional expressions. These findings were supported by the lexical diversity calculated between human and computer-generated reviews, where the former contained higher lexical diversity than the latter. These distinctions and linguistic patterns are vital pinpoints that may be used for discerning between genuine, human, and artificial, computer-generated content in the realm of fake review detection.

This research also comprehensively evaluated various AI models, encompassing ML, DL, and EL, to differentiate between human-generated and computer-generated Arabic reviews, aiming to propose an enhanced model for fake review detection. The results revealed diverse accuracies across individual ML models, with LR achieving the highest accuracy of 86.85%. However, the ensemble learning method on ML models exhibited a slightly lower accuracy of 83.19% compared to LR's accuracy. A reasonable reason would be that in soft voting, each classifier in the ensemble assigns a probability to each class, and the class with the highest aggregated probability is chosen. When a low-performance classifier frequently makes incorrect predictions, in this case, the DT produced a low accuracy of 70.08% compared to the other ML models, it adds to the pool of incorrect probabilities, potentially leading the ensemble to make erroneous decisions. This might explain why the EL model produces lower accuracy than LR.

Conversely, DL models demonstrated higher accuracies, with LSTM, CNN, and RNN achieving accuracies of 88.15%, 88.55%, and 86.88%, respectively. The application of the

EL method to DL models resulted in a slight performance improvement of only 1.02%, yielding an accuracy of 89.57%. A plausible explanation is that the models' probabilities are high and consistent, leading the ensemble learning model to achieve high accuracy.

Furthermore, our study delved into hybrid techniques, specifically combining the top-performing ML and DL models (LR and CNN). This hybrid approach achieved an accuracy of 89.70%, which is notably close to the performance of the leading transformer-based model, AraBERT, at 90%. The hybrid model, composed of CNN and LR, is easier to interpret locally, focusing on specific input features or patterns (*Springenberg et al., 2014*; *Zeiler & Fergus, 2014*). This simplicity helps understand the model's decisions and feature contributions, contrasting with AraBERT, which captures global context but relies on complex relationships among multiple words (*Vig, Dorr & Shneiderman, 2019*). This complexity may not suit certain text classification tasks, such as the one in this study.

In summary, this study aimed to explore diverse AI approaches to address the complex challenge of distinguishing between human and computer-generated Arabic reviews and shed light on the linguistic and emotional characteristics inherent in authentic and AI-generated textual data. Unlike previous studies that focused mainly on specific ML, DL, and transformer-based models, our study delves into EL and hybrid ensemble techniques. By combining the strengths of ML and DL models, we achieved accuracies comparable to individual DL models and EL methods applied to DL models and on par with TF models.

## LIMITATIONS AND FUTURE WORKS

Although this work has shown encouraging results and made valuable contributions in distinguishing between text originated by humans and text generated *via* computers, it is important to recognize that several limitations need to be addressed. These constraints emphasize the need for further investigation and advise prudence when interpreting the results. While comprehensive, the dataset used in the research may not completely include the wide range of writing styles and genres seen in human-written texts. Furthermore, the examined reviews were written in Arabic and obtained from a restricted set of venues that provide product reviews. The topic of product reviews could have had an impact on the variety of language used in the reviews. Future research should include more heterogeneous datasets, including a wider array of genres.

Another limitation is that the research was conducted by compiling ML, DL, EL, and TF models with limited feature engineering approaches. This may not fully include a variety of contextual factors that may be influential in detecting nuances between human and computer-generated reviews. Even though computer models might not be able to fully replicate the cultural, social, and contextual details like humans, their evolving intelligence of imitating human writings might lead to the need to develop advanced models trained on contextual content. This necessitates the compilation of increasingly sophisticated models using a range of feature engineering techniques, such as LLMs word embeddings, and distinctive textual features.

## CONCLUSIONS

The spread of fraudulent reviews has escalated in recent times owing to the enhanced accessibility and cost-effectiveness facilitated by advancements in LLMs such as ChatGPT. This study focus specifically on online Arabic reviews, where the intricacies of the Arabic language pose unique challenges for analysis. The study investigated ML, DL, TF, and an ensemble learning approach to develop a model capable of distinguishing between human-authored text and computer-generated text, such as that produced by ChatGPT. Additionally, the study explored linguistic disparities between human and GPT-generated text across three word use categories: POS, emotions, and linguistics. The findings highlight the efficacy of leveraging a combination of hybrid ensemble LR and CNN models for detecting human and computer-generated reviews, achieving an accuracy of 89.75%, which is comparable to the performance of the transformer-based model AraBERT, which achieves an accuracy of 90.0%. This demonstrates the potential of ensemble learning methods as a robust alternative to TF models in the context of Arabic text analysis.

Moreover, the results of the textual analysis provided insights into the linguistic differences between human and computer-generated reviews in an effort to detect false online customer reviews. Human reviews used more conjunctions, adverbs, hedges, intensifiers, and emotional words, whereas computer-generated reviews contained higher frequencies of nouns, proper nouns, verbs, adjectives, and assurances, with a notable increase in expressions of disgust. The study could serve as a benchmark for academics developing models to detect fraudulent reviews using advanced ML, DL, and EL approaches. Future research should refine EL techniques, such as stacking methods, and employ transformer-based models, exploring additional linguistic and contextual features to enhance the accuracy and reliability of fake review detection systems in real-world applications.

### Funding
This work was supported by Princess Nourah bint Abdulrahman University Researchers Supporting Project number (PNURSP2024R719), Princess Nourah bint Abdulrahman University, Riyadh, Saudi Arabia. The funders had no role in study design, data collection and analysis, decision to publish, or preparation of the manuscript.

### Grant Disclosures
The following grant information was disclosed by the authors:
Princess Nourah bint Abdulrahman University: PNURSP2024R719.
Princess Nourah bint Abdulrahman University, Riyadh, Saudi Arabia.

### Competing Interests
The authors declare that they have no competing interests.

## Author Contributions

- Fatimah Alhayan conceived and designed the experiments, performed the experiments, analyzed the data, performed the computation work, prepared figures and/or tables, authored or reviewed drafts of the article, and approved the final draft.
- Hanen Himdi conceived and designed the experiments, performed the experiments, analyzed the data, performed the computation work, prepared figures and/or tables, authored or reviewed drafts of the article, and approved the final draft.

## Data Availability

The third-party data was provided for peer review.

Access to the dataset must be requested from the dataset owners, the authors of the article "Detecting Arabic Fake Reviews in E-commerce Platforms Using Machine and Deep Learning Approaches": Samaher Alharthi, samaher.alorabi@gmail.com; Rawdhah Siddiq, rosiddiq@stu.kau.edu.sa; Hanan Alghamdi, hsaalghamdi@kau.edu.sa.

## Supplemental Information

Supplemental information for this article can be found online at http://dx.doi.org/10.7717/peerj-cs.2345#supplemental-information.

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
