# Peer review of "Ensemble learning approach for distinguishing human and computer-generated Arabic reviews"

_PeerJ Computer Science, doi:10.7717/peerj-cs.2345_

## Round 0.1 · original submission · Major Revisions

After carefully considering the reviews and assessing your manuscript, I am pleased to inform you that we would like to invite you to revise and resubmit your manuscript for further consideration. The reviewers have provided constructive comments that will help strengthen your work. Please address each of these points thoroughly in your revised manuscript. Additionally, ensure that you provide a detailed response letter outlining how you have addressed each comment raised by the reviewers. This will help the reviewers and myself to evaluate the changes made to the manuscript.

Also address few of my comments:
1) The term "Human or computer? " in the title is not adding any meaning, so better you drop it.
2) Compare you method with recent studies.

Reviewer 1 ·

Basic reporting

uthors have proposed a study to classify human and computer-generated Arabic reviews, with the aim of detecting fake reviews produced by computers.
This is an interesting study ,however there are some points that need to addressed strictly before decising about its acceptance. May address the following points.

Experimental design

Clarify the methodology section by providing more details on the feature selection process
Provide more context on the limitations of the study
In the results section, provide additional statistical analysis to support the observed differences in lexical and emotional features between human-generated and machine-generated reviews

Validity of the findings

Comparison of the proposed approach with existing methods for fake review detections required
Incorporate relevant references to recent studies in the field of natural language processing and fake review detection to provide a comprehensive review of the existing literature.
Provide recommendations for future research directions

Additional comments

English needs improvements at various places

Reviewer 2 ·

Basic reporting

The level of english should be improved in the revised version.
There is no numbering is used in the refrences section. it should be modified in the revised version. The refrence number in the text and the refrences should be in number format.
I dont think so they followed journal template for the figures and the tables. Plesae take a look journal template.
The quality of figures in not satisfiied. it should be improved and in HD form in the revised version. few mathimatical equation has no numbering. Plesae check carefully in the manuscript.

Experimental design

The paper in scope of this journal.
In mostly equations used in the manuscript needs proper expination in text and should be properly cited in the proper place in text.
The problem statment is not clear. it should be mention in the abstract section. The implication of this research should be also mentioned.
1- There is unnecessary information in the abstract, thus, it should be polished. We expect a scientific manuscript presenting the results of the study, not a report summarizing a study. I think we should see the following information in a few sentences in the abstract.
1.1. What was done in this study?
1.2. Why was this study done?
1.3. What are, the implications of this study?
1.4 The achived results should be the part of abstract section.

The intrdouction section is very short. it lacks motivation, contributation and the benfits of this research. You should mention problem statment, state of art and the few drawbacks in this section. current contributation is not clear it should be polished.
Draw a comparsion table in section 2 and compare most recent studies along with advanatges and drawbacks.

Validity of the findings

Section 3 should be the part of section 1.
in section 4 please mention simulation enviroment, tool and the parametrs.
The results sectuion needs significant improvment. what figure 4 tells us? what are the x-axis and Y-axis means? it seems 4 diffrenet methods compared. I did not find your own proposed methods. plesae write your proposed method and tell how it perform well? from current graph, it seems all were similer.
one results is not enough. more experiments should be performed and the results clearly written in the text and the graphs.
I check the dataset. it is not in arabic. The text is not readable format.
The conclsuion section should be modified by adding key findings.

---

## Round 0.2 · accepted · Accept

I am pleased to inform you that your paper has been accepted for publication in PeerJ Computer Science. Your research makes a significant contribution to the field, and we believe it will be of great interest to our readership. On behalf of the editorial board, I extend our warmest congratulations to you.

Novelty (as per the comments from R2) is not one of the criteria at PeerJ

Reviewer 1 ·

Basic reporting

Seems ok now.
English still needs improvements

Experimental design

no comment

Validity of the findings

no comment

Reviewer 2 ·

Basic reporting

I did not see a clear improvement in this paper. It has a lack of novelty.

Experimental design

.

Validity of the findings

.